# Information and Public Knowledge of the Potential of Alternative Energies

Galvão Meirinhos [1], Mariano Malebo [2], António Cardoso [2], Rui Silva [3,4,*] and Reiville Rêgo [5]

1   LABCOM-IFP, University of Trás-os-Montes e Alto Douro—UTAD, 5001-801 Vila Real, Portugal; galvaomeirinhos@gmail.com

2   Department of Business and Communication Sciences (DBCS), University of Fernando Pessoa—UFP, Praça 9 de Abril 349, 4249-004 Porto, Portugal; 37418@ufp.edu.pt (M.M.); ajcaro@ufp.edu.pt (A.C.)

3   CETRAD Research Center, University of Trás-os-Montes e Alto Douro—UTAD, 5000-801 Vila Real, Portugal

4   NECE-Research Center in Business Sciences, University of Beira Interior, 6201-001 Covilha, Portugal

5   Campus Tomé-Açu, Universidade Federal Rural da Amazônia/UFRA, Rod. PA 140, 2428-4822, Tomé-Açu 68680-000, PA, Brazil; reiville.rego@ufra.edu.br

\*   Correspondence: ruisilva@utad.pt

**Abstract:** The objective of this research project is to study the economic development model of the Angolan economy in order to analyze the adoption of an alternative strategy capable of leveraging the economy, based essentially on alternative energies, and therefore, to demonstrate and prove the need to diversify Angola's economic model, highlighting the benefits of a diversified versus a non-diversified economy with respect to sustainability. The first stage of the design of this empirical study involved establishing a focus group in order to construct and adjust a data collection instrument in the form of a questionnaire to be applied to a broader set of managers and informed professionals with a critical view of the country's future and the models and alternatives to economic development and diversification of the economy on a sustainable basis. Energy plays a fundamental role in Angola's economic and social development. Excessive dependency on the oil sector and inefficient production due to high costs, combined with changes in global environmental and energy policies, make it essential to reflect on the evolution of the country's energy sector, equating a different economic development model, the diversification of the economy, and the exploration of other sources of energy, such as biofuels. Renewable energies emerge as a safe, healthy, environmentally friendly and economically viable energy alternative that could bring the Angolan economy closer to that of developed countries. Biofuels have become popular and have begun to be seen as a valid alternative to fossil fuels because they have lower production costs and they cause less impact on nature. Furthermore, since they are biodegradable, they can be commercialized at a lower cost from renewable sources. According to the respondents, the research results show that the best energy alternatives to reduce oil dependency are solar energy, biodiesel, hydraulic energy, and bioethanol. An assessment of the attractiveness and potential of biofuels show that the best alternative is bioethanol, followed by biodiesel.

**Keywords:** Angolan economy; diversification; strategic alternative; biofuels

## 1. Introduction

The primary source of revenue for Angola's GDP is oil, thus, defining Angola's economic model as practically monolithic. Since oil is a commodity, a change in market value is predictable and can be positive or negative for Angolan interests. If the change in market value is negative, the Angolan government would be forced, as it has been in the past, to adjust its general budget to reflect updated oil prices. Oil is the country's largest generator of revenue, which could mean that many of the actions planned for a given year would not be carried out. This situation could force expenditure restraint or restriction situations and could even generate serious social and political pressures.

The World Bank's Report No. AUS6794, clearly stated that "an effective economic diversification strategy could increase Angola's long-term GDP growth trajectory". In the report, there is an obvious association between economic growth and the need to diversify domestic production, and therefore, create a more balanced fiscal balance. Based on the above, we justify the need to address this topic, since it is clear that strategic alternatives for Angola are necessary and mandatory to concretely change their economic model, generating greater sustainability and economic growth in the long term.

According to the Central Intelligence Agency, the Angolan economy is driven by the oil sector, representing around 50% of the GDP, accounting for more than 70% of the government's revenues, and corresponding to more than 90% of the country's exports, which confirm Angola's flagrant dependency on the oil sector. Oil prices are defined on the international markets with daily price oscillations, which become an economic instability factor and a problem for managing an economy that depends essentially on the oil industry. On the one hand, the problem of the Angolan economy lies precisely in the fact that oil is a non-renewable natural resource that could compromise the country's economic position in the long term. On the other hand, as we have seen, oil is a commodity and the Angolan government has no control over its price since it depends on the international markets. This reality already occurred in 2008, 2009, 2015, 2016, and 2017, when the Angolan government's general budget had to be revised due to a drop in the price of oil on the international markets. Between July and December of 2008, oil prices fell by about 70%, and since then, have fluctuated constantly. According to the International Monetary Fund Report No. 18/157, Angola, although it is the second largest oil producer in Africa, suffers severely when the price of oil is between USD 50 and 55, which involves reducing oil production because it is unprofitable, thus, creating severe budgetary problems because of the gigantic investment needs in infrastructure and social spending. The ideas presented in the abovementioned report justify the need for this research on energy alternatives. Therefore, we aim to analyze strategic alternatives for diversification of the Angolan economy and to evaluate the attractiveness of biofuels.

To better frame the knowledge problem and objective, this research was conducted in three phases:

1.  In the exploratory phase, we conducted a bibliographical review to obtain greater familiarity with the theme's problem, to clarify the understanding and focus of the research on alternative energies and, to facilitate the construction of the research hypotheses [1].
2.  In a second phase, we collected the opinions of Angolan economic experts on the sustainability of strategic alternative energies for diversification of the Angolan economy. To support the analysis, the Delphi technique was adopted to obtain a specialized understanding regarding the sustainability of ethanol or other energy possibilities, as possible strategic alternatives for diversification of the Angolan economy.
3.  In the third phase, we applied a debugged and validated survey to understand, from the point of view of managers and specialists from various activity sectors, the possible energy alternatives as a means of diversifying the Angolan economy.

This is an exploratory study with a descriptive design in which, in the first phase, we established a "focus group" to construct and adjust a data collection instrument in the form of a questionnaire to be subsequently applied to a broader set of managers and specialists [2,3].

Despite the vast literature that addresses the phenomenon of economic development, and the related areas of knowledge, the approaches have been focused on territorial scales. Thus, it is important to address the exogenous and endogenous economic development models that are relevant to the configuration of an intended analysis. According to [4] endogenous economic growth is long-term growth determined by internal forces in the economic system. Development is based, though not exclusively, on locally available resources, local knowledge, culture, and leadership. It has mechanisms for local learning and experimentation, building local economies, and retaining benefits in the local area [5,6].



## 2. Literature Review

### 2.1. Models of Endogenous Development of an Economy

Endogenous development is a paradigm based on the basic idea that the productive system of a country grows and is transformed using the development potential existing in the territories, that is, regions and cities, through investments made by companies and public entities, under the control of local communities, with the ultimate goal of improving the standard of living of the populations in these territories [7,8]. In this sense, it is clear that the concept of endogenous development integrates the social and economic dimensions. The protagonism claimed for the territorial dimension, in turn, is suggested not only as an expression of the spatial anchorage of organizational and technological processes but, equally, of the circumstance that any locality or region offers itself as the result of a history that has been shaping its economic, cultural, and institutional environment.

Endogenous development is linked to the dynamics of a country, its cities and regions and the network of agents and interests that give them substance. This is to underline, in line with what [9–11] have stated, among others, that the processes of growth and structural transformation that take place arise as a consequence of the transfer of resources from traditional to modern activities, the exploitation of external economies, and the introduction of innovations, which are aimed at increasing the well-being of the population of the city, locality, or region that generated a change. In other words, growth is organized around the expansion and transformation of pre-existing activities, using the resources and innovation potential available in a territory, conditioned by the social and cultural structure and codes of conduct of human communities based in particular spaces, which favour or limit it and, in any case, give it its unique shape.

From a policy point of view, starting from the framework described above, the actions to be developed should take into consideration the availability of the country's resources and promote their economic enhancement, whether natural resources or others. The solidity of the economic affirmation processes and the capacity to internalize the wealth generated is due to political initiatives that should take advantage of the network of local solidarities and the existing or developed concertation capacity, involving all the economic operators' social agents and political decision-makers. The emphasis on a country's potential, which is the starting point for this approach to development, takes the form of a policy to enhance the resources and capacities of a region or country, which, as we shall see below, must be at the root of regional or even national policymaking.

### 2.2. Models of Exogenous Development of an Economy

An exogenous model studies the growth of a country's economy over a long period of time. The model presents the source of economic growth: capital accumulation, labor force growth, and technological change [12]. was concerned with demonstrating that product per capita was an increasing function of the relationship between capital and labor. Labor force grows at a natural rate (exogenous to a model). In this sense, an amount of savings per capita is necessary, which must be used to equip new workers with capital per capita equal to that of other workers. The other part of the savings should be used to guarantee non-depreciation of the capital. The first part of the savings quoted above to equip new workers is called "capital enlargement" (expansion of the labor force), and the savings used to increase the capital/labor ratio is called "capital deepening". To reach a steady-state situation, the savings per capita must equal capital enlargement. The capital per worker has a decreasing income, therefore, when this equilibrium point is reached, there is no point in investing more in a worker who has per capita savings equal to the capital expansion because this worker's productivity will not be maximized. Thus, the conditioning factor of economic growth is the growth rate of the labor force.

For [7], exogenous economic growth is long-term growth determined by forces that are external to the economic system. Exogenous development restricts the use of endogenous resources. It seeks opportunities for economic development abroad, considering the supply of raw materials, as well as knowledge, financing, skilled labor, and markets [5,13]. In the

case of the Angolan economy, it is necessary to analyze the conditions and potentialities that the country possesses and to evaluate which economic development model best adapts to its reality. However, determining the optimal size of the public sector is difficult. The state's concern with maximizing long-term growth must weigh the effects of public intervention policy and the growth-retarding effects of higher taxes and regulations.

Regarding higher taxes and regulations, economic growth theory that takes consideraiton public sector functions such as correcting market failures, investments in infrastructure, and taxes, may neglect the state's role in redistributing income and how policy behavior is determined by sometimes conflicting interests. This is a situation that occurs in many countries, and we believe that sometimes political decision-makers are not in a position to make strategic decisions for the development of their countries.

### 2.3. Oil

Brazil's National Electrical Energy Agency [14] defines oil as a flammable oil formed, over millions of years, from the decomposition of organic matter such as plants, marine animals, and vegetation typical of flooded regions and found only in sedimentary terrain. Oil is composed of hydrocarbons, made up of carbon and hydrogen, to which atoms of oxygen, nitrogen, and sulphur can be added, as well as metallic ions, mainly nickel and vanadium. According to BP, world oil production in 2014 was around 4226.60 million tons per day, while daily consumption was around 4211.10 million tons. From the figures presented at the time, one can see that oil is a resource in high demand on the international markets [15]. Oil is a primary source resource on the stock exchange and its price is determined by supply and demand. Therefore, this explains the importance of oil when analyzing some of its derivatives, such as petrol, paraffin, diesel, asphalt, synthetic rubber, lubricants, and plastics, among others. The BP report published in June 2015 illustrated the importance of oil in the Angolan economy, as the second largest oil producer on the African continent, with the ranking led by Nigeria [16].

The Economic Report of Angola 2016, prepared by the Catholic University of Angola, outlined the importance and impact of oil, evidencing the clear dependency on oil in relation to the economy. Oil is undoubtedly of paramount importance to the Angolan economy because it leverages a good part of other sectors of national activity such as agriculture, fisheries, manufacturing, and transport. Thus, economic sustainability involves a greater balance in the sources that generate the gross domestic product to generate new opportunities and reduce costs by reducing the need to import and leverage new possibilities and opportunities for internal capacity building and investments in new forms of energy production. When nearly 30% of the state budget is dependent on oil and gas revenues, the country is in a weak position to make reforms and strategic investments for the integral and ongoing development of the Angolan economy and society [17].

### 2.4. Biofuels

According to [18], biofuels are obtained from renewable organic matter, also called biomass, which can be products of animal or vegetable origin, as is the case of sugar cane, corn, soya, sunflower seeds, wood, and cellulose. Therefore, it is possible to produce fuels such as alcohol, ethanol, or biodiesel from these products. Biofuels are popular because they are a valid alternative to fossil fuels such as oil in specific sectors. In addition, they have lower production costs because they cause less impact on nature since they are biodegradable, they are marketed at a lower cost, and they are the result of renewable sources.

Brazil is an example of one country that have been looking at biofuels since very early on. According to the Ministry of Mines and Energy (MME), the pioneer tests were carried out between 1905 and 1925. In 1931, the Brazilian government established a decree that made mixing 5% alcohol in imported gasoline compulsory. However, with the discovery of extensive oil reserves in the Middle East, interest in biofuels declined globally. However, with the first world oil crisis in 1973, the search for new energy sources re-emerged.

In 2015, Paris, France hosted the 21st United Nations Conference on Climate Change (COP 21), whose objective was to bring countries to an agreement on global warming by reducing the emission of greenhouse gases [19]. Unfortunately, that same year, there was a drastic fall in oil prices, generating considerable constraints in the economies of the producing countries, which had to review their budgets and were forced to reflect on alternatives. However, even before the great discussions on climate, greenhouse effect, and oil crisis, several countries were already producing biofuels in considerable quantities, such as the United States of America, Brazil, Germany, and Indonesia.

*2.5. Ethanol*

Ethanol is obtained from sugar cane as a biofuel, since the term biofuel is generic and may encompass several types and several origins. Nevertheless, we identified different energy alternatives in the questionnaire, such as biodiesel, algae biodiesel, H-BIO, geothermal, hydraulic, solar, wind, and tidal energy alternatives. According to Petrobras, ethanol is alcohol with an oxygenated organic compound, also called ethyl alcohol, and its chemical formula is $C_2H_5OH$. Ethanol is obtained from various raw materials such as sugarcane, corn, manioc, and sugar beet [18].

According to the Ethanol Industry Association (IEA), and in terms of applications and uses of ethanol, it can be used as a raw material in three areas, i.e., beverages, fuels, and industry, the latter being the final use in: the manufacture of pharmaceuticals, cosmetics, toiletries, detergents and cleaning products, printer ink cartridges, paints, and coatings. The ethanol industry will always depend, in a first analysis, on the existence of the minimum conditions for the generation of raw material, and the potential that this raw material has in terms of quantities and respective renewal conditions [20]. According to [21], implementing biofuel projects in Angola require about two years of research related to product tests and choices of best species. In addition to expenses and direct costs required to implement an ethanol industry, there are other important areas to take into consideration, such as legislation, equipment, and human capital that always vary, and therefore, require a feasibility study involving experts in the most distinct areas, who can determine with proven evidence the probability of success or otherwise in implementing an ethanol industry.

Concerning the ethanol industry, the approach taken in this study is not focused on the details of the feasibility or otherwise of implementing an industry. Rather, the focus is on an analysis that seeks to understand the capacity these industries can offer to Angola's GDP in terms of economic and social impact, based on existing successful experiences with already implemented industries. An economic development model of the Angolan economy is highly dependent on its endogenous characteristics, namely its oil production capacity, but it is also totally dependent on exogenous characteristics, namely the international fixing of oil barrel prices. This circumstance leaves the country's real economic growth dependent on finite natural resources and the impossibility of intervening in the fixation of the price of a barrel of oil. In this sense, there is a possibility that alternative energies may be a solution to the abovementioned dependency on oil, namely through ethanol production.

## 3. Methodology

The definition of the research problem, the research questions, and the objectives were crucial to the definition of the methodological and operational framework since the aim was to analyze the perception of specialists on alternatives for diversifying the economy and the viability of biofuels as a way of breaking the dependency on the Angolan oil industry. This empirical study was developed, in the first stage, by setting up a focus group in order to construct and adjust a data collection instrument in the form of a questionnaire to be applied to a broader set of managers and specialists. The methodology used in this study is described in detail below.

Given that the nature of the research problem focuses on understanding the social and economic phenomena relative to Angola's development strategies and economic diversification, as well as the evaluation of energy alternatives that seek to reduce dependency on the

oil industry, equating the potential of biofuels, in this study, we began by considering the spatial, cultural, and organizational context in which the phenomena occurred within the framework of the naturalistic, exploratory, and descriptive nature of the study [2,3]. Thus, the previous literature review provided information that allowed us to know the "state of the art" and to support the empirical research model, contributing to the development of the methodological path. Furthermore, the process of constructin the data collection models and instruments took place in close inter-relation and cooperation with the perspectives of expert actors since they knew the terrain and the socio-cultural, economic, technological, and political realities where the phenomena occurred [22,23]. Thus, first, it was deemed appropriate to conduct an exploratory approach through a focus group (qualitative analysis) with some experts of different nationalities (Angolan, Portuguese, and Brazilian) in order to identify practices, models, strategies, interests, and problems that could serve as a basis for the development of a questionnaire (quantitative analysis) to be applied to a broader and more robust sample of experts.

The research problem was translated into the following initial questions: (1) What are the strategic alternatives for diversification of the Angolan economy? (2) What is the potential/attractiveness of biofuels to reduce dependency on the oil sector? Based on these starting questions, many other questions arose, namely:

- What are the experts' perceptions of the strategic goals for sustainable development and competitiveness?
- What are Angola's main economic, social, and environmental vulnerabilities?
- What is the impact of clusters on the development of the economy and business competitiveness?
- What are the priority goals for Angola's economic development?
- What is the time frame for the development of an alternative diversification model?
- What are the priority sectors to develop in Angola?
- What are the best government measures to diversify the Angolan economy?
- What are the strategic alternatives for diversification of the economy?
- What are the reasons for the lack of competitiveness in the oil sector in Angola?
- What are the energy alternatives to oil to be explored in Angola?
- What is the potential/attractiveness of biofuels?

The methodology adopted in this research required the identification of specialists to be involved in the study in order to generate qualified information and to participate interactively in the various phases, and then to explore the information obtained, generating consensus or guidelines, either by validating the information produced or by the experience developed in the form of benchmarking. Thus, the principles inherent to Delphi techniques were followed, following [24–28].

As previously mentioned, the first phase of the questionnaire required the interpretation of the models and theoretical concepts for the identification and classification of the dimensions and explanatory variables; 11 dimensions and 56 variables were identified (Table 1):

**Table 1.** Identification of dimensions and variables.

| Dimension | Most Important Variables |
|---|---|
| Strategic goals for sustainable development and competitiveness | Diversification of Angola's economy |
| | Reducing external dependency |
| | Decrease in dependency on the oil sector |
| | Development and empowerment of human resources (science, education and training) |
| | Development of infrastructures (communications and transportation) |
| | Creation of a local environment that promotes private investment and the attraction of foreign investment |

**Table 1.** *Cont.*

| Dimension | Most Important Variables |
| --- | --- |
| Angola's economic, social, and environmental vulnerabilities | Bureaucracy<br>Corruption<br>The informality of the economy<br>Excessive weight of the state in the economy<br>Transport and communication infrastructure costs<br>External dependency<br>Dependency on the oil sector |
| Clusters in economic development and business competitiveness | Promotion and development of different sectors (agriculture, industry, and new services) can make the country more competitive |
| Priority targets for Angola's economic development | Valorization of human capital (education, capacity building, and training)<br>Valorization of "endogenous" resources<br>Reduction in external dependency<br>Increase in internal productivity<br>Promotion of exports |
| Time horizon | From 5 to 10 years<br>From 10 to 20 years<br>From 20 to 30 years<br>From 30 to 40 years |
| Priority sectors to be developed in Angola | Primary sector (agriculture, fisheries, and forestry)<br>Secondary sector (development of industry)<br>Tourism<br>Energy<br>Commerce<br>Construction<br>Services<br>Education |
| Government measures to diversify the Angolan economy | Economic policies<br>Fiscal policies<br>Education policies<br>Support for innovation, science, and technology. |
| Strategic alternatives for diversification of the economy | Valorization of human resources<br>Valorization and exploitation of endogenous resources Science, innovation, and technology<br>Reduction in imports<br>Increase in exports |
| Lack of competitiveness in the oil sector | The complexity of the sector<br>Price dependency on international markets<br>Lack of seriousness and delays in accountability |
| Energy alternatives to oil | Clean energies to consider: solar, biodiesel, bioethanol, geothermal, hydro, wind, marine, algae biodiesel, H-bio |
| Potential and attractiveness of biofuels | Evaluation criteria: technology costs, gas emissions, productivity, natural resources, contribution to competitiveness, energy potential, and systemic innovation |

After identifying the concepts associated (dimensions and variables) with the diversification of the economy and the evaluation of strategic alternatives for growth, these were assessed by a group of specialists connected to the economic and energy sector and with knowledge of the Angolan reality. Each participant was given a summary of the concepts and a pre-questionnaire for evaluation during the focus group. The session and group discussion took place in a conference room equipped with multimedia support and recorded on video and audio support [29]. The focus group discussions included the

following: (1) participants expressed their ideas in a free "open-minded" way; (2) participants expressed their opinions about certain keywords associated with the concepts under study; (3) participants evaluated the research questionnaire proposal. The focus group was composed of eight members with complementary skills: two economists (both were university professors), two petroleum engineers, two public managers, and two consultants in economic development with experience and work carried out at the UN.

The focus group participants generally considered it urgent that Angola diversify its economy and invest in a new model of economic development based on the potential of its endogenous resources, therefore, leading to reduced imports (reduce dependency on foreigners) and increased exports. They considered the importance of a new dynamic of private business activity (to reduce the weight of the state) based on entrepreneurship, knowledge (education and training of human resources), and innovation and technology (infrastructure, transportation, and communications). This activity would industrialize the country, and the state should create the conditions and policies (economic, financial/fiscal, social, and technological) necessary for its development.

The specialists and professionals involved in this phase also contributed by selecting or adding variables corresponding to the dimensions defined in the initial questionnaire. For each dimension, the experts expressed their opinions, suggesting the elimination or addition of concepts and/or variables. The focus group participants were unanimous in affirming that the "strategic goals for sustainable economic development", conceived for the medium and long term, should include diversification of Angola's economy and a reduction in foreign dependency and the oil sector. They also indicated the need for the development and training of human resources (science, education, and training), the development of infrastructure (communications and transportation), and the creation of a local environment that promotes private investment and attracts foreign investment.

Concerning the vulnerability dimension, the professionals recognized the bureaucracy, corruption, and informality in the Angolan economy. They also noted the excessive weight of the state in the economy, the costs of transport and communications infrastructures, Angola's foreign dependency, and the oil sector. The creation of several clusters could promote and develop different sectors in Angola related to agriculture, industry, and new services, making the country more competitive. The following priority goals were noted: the "valorization of human capital" (education, capacity building, and training) and of "endogenous resources"; reduction in external dependency; and increased internal productivity to promote exports. The speakers were not unanimous as to the period needed to implement policies and programs to diversify the Angolan economy, with time indications ranging from 5 to 40 years, above all, because they considered that there were medium-term initiatives (economic and fiscal programs) and other long-term innitiatives (science and technology).

Regarding the priority sectors, the specialists considered it relevant to focus on the primary sectors (agriculture, fishing, and forestry) and the secondary sector (development of industry) to valorize the country's immense exogenous resources, and thus, reduce economic dependency on the exterior. However, they also indicated tourism, energy, commerce, civil construction, services, and education. In terms of government policies, they pointed to fiscal and educational economic policies as priorities and support for innovation, science, and technology. In terms of strategic alternatives, the participants referred to "valorization of human resources", "valorization and use of endogenous resources", "science, innovation, and technology", and "reduction in imports and an increase in exports". The specialists stated that they believed the lack of competitiveness in Angola's oil sector was due to the complexity of the sector, the dependency on oil prices in the international market, and a lack of seriousness and delays in accountability. In terms of alternatives, the professionals indicated the feasibility of several alternatives considered to be sustainable and environmentally friendly, such as solar, biodiesel, bioethanol, geothermal, hydro, wind, tides, biodiesel from algae, and H-bio. Their choice should result from evaluating some

criteria, such as technology costs, gas emission, productivity, natural resources, contribution to competitiveness, energy potential, and systemic innovation.

The third phase of the process consisted of creating the questionnaire, which formulated statements or prepositions based on the research objectives, taking into consideration the dimensions and variables identified in the literature and the contributions identified through the focus group. A 5-point Likert-type ordinal scale was associated with each statement in the questionnaire, selected at random, ranging from "1 (I strongly disagree)" to "5 (I strongly agree)". In addition, some sociodemographic questions were also included to characterize the respondents, such as gender, age, education, marital status, activity sector, and nationality. This phase included validation of the questionnaire, which functioned as a pretest of the research instrument [3].

### 3.1. Sample

Given the theme to be explored in this dissertation, it was essential to obtain information from specialists and informed professionals with a critical view of the country's future and the models and alternatives to economic development and diversification of the economy on a sustainable basis. Furthermore, given the impossibility of investigating the entire population, using a sample was the most effective way to study and understand the phenomenon, generating empirical material for analysis [3]. Thus, a set of national and international experts was identified, and, after obtaining their e-mail addresses, the questionnaire was sent to each of the respondents in the set. Therefore, this was a non-probability convenience sample [30]. A total of 160 questionnaires were sent out, and 120 questionnaires were returned and validated by the deadline of 30 September 2020. In addition to the essential inclusion criterion that the respondent must be a specialist or professional in the areas of economics, management, politics, education, or engineering, the opportunity and availability criteria were also taken into consideration in order to participate in the production of information for a period of time, limited to two months, by completing and returning a questionnaire distributed via email.

### 3.2. Data Collection Tools

After the data collection instrument administration, the next stage of the research process was data analysis and interpretation to find answers to the research problem and objectives [31]. Given the exploratory nature and design, a univariate data analysis was performed using descriptive statistics (frequency, means, and standard deviation) [31,32]. The Statistical Package for Social Science (SPSS) version 25 [33] was used for data analysis.

The first part of the questionnaire contained questions for characterizing the respondents (gender, age, education, marital status, activity sector, and nationality), thus, constituting independent variables (pre-existing characteristics). In the second part of the questionnaire, explanatory variables were included referring to the dimensions subject to measurement in the form of variables, alternatives, and criteria (see Table 2).

**Table 2.** Dimensions and items used in the questionnaire.

| Dimension | Number of Items |
| --- | --- |
| Strategic goals for sustainable development and competitiveness | 14 variables |
| Economic, social, and environmental vulnerabilities of Angola | 20 variables |
| Clusters in the development of the economy and business competitiveness | 7 variables |
| Priority goals for Angola's economic development | 6 variables |
| Time horizon | 4 alternatives |
| Priority sectors to be developed | 8 variables |
| Government measures to diversify the economy | 9 variables |

**Table 2.** *Cont.*

| Dimension | Number of Items |
|---|---|
| Strategic alternatives for diversifying the economy | 10 variables |
| Lack of competitiveness in the oil sector | 5 variables |
| Energy alternatives to oil | 9 alternatives |
| Potential and attractiveness of biofuels | 8 criteria |

Source: Authors.

Considering the research problem and objectives, the construction of the questionnaire was based on the studies and works identified in the literature, as well as on the contributions obtained in the focus group.

To assess the "strategic goals for sustainable development", the work of [34] was taken into consideration, as well as endogenous development models [35–37] and the philosophy of sustainable development [38] that seek to integrate in a balanced way the economy, society, and natural environment [5,38]. Thus, taking into consideration the priority goals for sustainable development, the respondents expressed their degree of agreement with the statements presented (14 statements) on a 5-point Likert-type scale [30], ranging from "totally disagree" to "totally agree". The variables included were:

**V1** Improve the local business investment environment;
**V2** Invest in tangible strategic infrastructure;
**V3** Invest in business parks and facilities;
**V4** Invest in intangible strategic infrastructure;
**V5** Promote local business growth;
**V6** Promote the creation of new enterprises;
**V7** Attract foreign investment;
**V8** Develop business sectors and clusters;
**V9** Integrate unproductive or hard-to-employ workers;
**V10** Establish an adequate system of environmental protection, natural disaster prevention, and air and maritime safety;
**V11** Develop, train, and empower human resources;
**V12** Develop transportation and communications;
**V13** Promote science, technology, innovation, and entrepreneurship;
**V14** Diversify the economy.

Regarding the "economic, social and environmental vulnerabilities of Angola", the indications of the [38] were used as a reference, subsequently adapted by [5,39]. Respondents expressed their degree of agreement with the statements presented (20 prepositions) on a 5-point Likert-type scale [30], ranging from "totally disagree" to "totally agree". The selected statements were as follows:

**V1** A narrow resource base and little or no opportunity to create economies of scale;
**V2** Small domestic markets, heavy dependency on some external markets, and long distances from export and import markets for resources;
**V3** High energy, infrastructure, transport, communication, and maintenance costs;
**V4** Low and irregular international traffic volumes;
**V5** Fragile natural environments and vulnerability to natural disasters;
**V6** Small but growing population;
**V7** High volatility of economic growth;
**V8** Limited opportunities for the private sector;
**V9** A proportionately large dependency of the economy on its public sector;
**V10** A disproportionately costly public administration;
**V11** Corruption and informality of the economy;
**V12** Restricted access to credit;
**V13** Deficient energy distribution systems;

**V14** High inflation;
**V15** Excessive bureaucracy;
**V16** Inefficient judicial system;
**V17** Unskilled labor force;
**V18** Dependency on oil;
**V19** Weak currency;
**V20** Literacy/education of the population.

In order to assess the impact of "clusters on the development of the economy and business competitiveness" [40,41], works were used as a reference, which contemplates 7 statements on which respondents expressed their degree of agreement on a 5-point Likert-type scale [30] ranging from "strongly disagree" to "strongly agree":

**V1** Promote competitiveness of enterprises and locations;
**V2** Promote increased productivity of enterprises;
**V3** Facilitate complementarities between the activities of the different actors;
**V4** Facilitate access to institutions and benefits;
**V5** Help measure the performance of domestic activities and limit;
**V6** Opportunistic behavior;
**V7** Facilitate the implementation of innovations;
**V8** Facilitate the formation of new companies.

To assess the "priority goals for the economic development of Angola", the works of [5,39] were taken into consideration. After the specialists' contributions to the "focus group", six variables were selected on which the respondents expressed their degree of agreement. Namely:

**V1** Give sufficient focus to technological, innovation, and creativity systems as part of a sustainable development strategy;
**V2** Emphasize building of human capital through investments in education and training;
**V3** Give sufficient attention to the development or adoption of climate change resilience systems;
**V4** Address Angola's most critical issues, such as high debt levels, inadequate access to technology, difficulties with business transactions, and inadequate access to sources of finance;
**V5** Present energy alternatives to break the dependency on oil, namely in terms of biofuels;
**V6** Increase the rate of independency and reduce the imbalances of all factors.

In as much as the time horizon (period of time) that respondents deemed acceptable/realistic to develop an alternative model for the diversification strategy of Angola's economic development is concerned, the alternatives suggested by the professionals who participated in the focus group were taken into consideration:

-      From 5 to 10 years;
-      From 10 to 20 years;
-      From 20 to 30 years;
-      From 30 to 40 years.

To identify the priority sectors for Angola's development, the recommendations of the professionals involved in the focus group and the works of [42,43] were considered. The respondents put the recommendations in order of priority/importance according to "1" represents the highest priority" and "8" represents the least priority/importance. The sectors considered were: agriculture, livestock and forestry; tourism; oil and gas; manufacturing; diamonds and precious stones; construction; trade and distribution and the services, with the chance for respondents to include and indicate "other sectors".

In order to assess the "government measures to diversify the economy", the work of [39] was taken into consideration. After the "focus group", 9 statements/positions were contemplated by the respondents who indicated their degree of agreement:

**V1** Tax reduction;
**V2** Subsidized interest rates;
**V3** Economic policies;
**V4** Actions to enhance human capital;
**V5** Export subsidies;
**V6** Creation of agendas for diversification and national agencies with responsibility for stimulating and coordinating the process of structural change;
**V7** Organize meetings, lectures, seminars, workshops, etc., to inculcate a new spirit of greater openness to international competition;
**V8** Technological innovation;
**V9** Tax systems and financial incentives that encourage diversification and stimulate private investment.

Regarding the evaluation of strategic alternatives for diversification of the economy, the works of [39,44] were considered, as well as the suggestions of the professionals who participated in the focus group:

**V1** Import substitution (through efficiency and not through administrative protection mechanisms, which only generate bureaucracy and corruption);
**V2** Production of intermediate products;
**V3** Valorization of national human resources (reducing dependency on expatriates);
**V4** Technological innovation;
**V5** Use of national raw materials;
**V6** Diversify export destinations;
**V7** Definition of long-term industrial and investment policies to promote the sustained growth of the Angolan economy;
**V8** Strengthening the role of the national investment system and development of the financial and banking sector;
**V9** Enhancing human potential and innovation;
**V10** The emergence of specializations around the strengthening of supply based on new producers and new services.

In order to assess the lack of competitiveness in the oil sector, the works of [45,46], and the report of the [47] were taken into consideration:

**V1** The revenues Sonangol receives from taxes and joint ventures and other sources of income do not appear in government accounts;
**V2** The price of oil is undervalued in the state budget, and any revenue above this estimate is never declared;
**V3** Government expenditure declarations are inaccurate;
**V4** The share of taxes and royalties that Sonangol pays to the government is transferred with significant delay and in local currency;
**V5** The network of financial arrangements created by oil-backed loans is complicated.

In terms of identifying and assessing energy alternatives to explore in Angola in order to reduce dependency on the oil industry [46], the suggestions raised by [48,49], and inputs from practitioners were followed. Thus, the following alternatives were included: biodiesel, bioethanol, biodiesel from algae, H-BIO, geothermal, hydraulic, solar, wind, and tidal.

Finally, and in order to evaluate the attractiveness of biofuels, we considered the works of [48,49] that, as considered by several experts [50,51], consider it pertinent to weight several criteria, through the use of the MACBETH method (decision support method that allows evaluating options taking into account multiple criteria) namely:

**Criterion 1** Cost of technology developed for production;
**Criterion 2** Emission of pollutant gases due to burning in engine combustion;
**Criterion 3** Job creation;
**Criterion 4** Productivity of raw materials;
**Criterion 5** Existence of natural resources;
**Criterion 6** Contribution to country's competitiveness;
**Criterion 7** Energy potential;
**Criterion 8** Innovation and systemic change.

Four biofuels were selected, and respondents were asked to rate each criterion based on the following scale: (1) extreme, (2) very strong, (3) strong, (4) moderate, (5) weak, (6) very weak, (7) null. The questionnaire was sent by e-mail to the specialists/professionals identified in the network of contacts which, after being completed, was returned for subsequent data analysis. This process took place in August and September 2020.

## 4. Analysis of Results

The sample was then characterized, followed by data analysis using descriptive statistics (absolute and relative frequencies, mean, and standard deviation), the assessment of the scale's reliability (Cronbach's alpha), and bivariate analysis of the data.

As can be seen in the following table, there is a gender imbalance among the respondents, with a higher percentage of females (86) than males (34).

The sample ranged in age from 30 to 65 years (Table 3). For operational reasons, three age groups were created. It is noted that the age group "up to 35 years old" consisted of 56 respondents (46.7%), followed by the age group "36–50 years old" (42 respondents), and then the age group "over 50 years old" with 22 respondents (9.4%).

**Table 3.** Sample.

| Title | Title | F | % |
|---|---|---|---|
| **Gender** | Male | 34 | 28.3 |
| | Female | 86 | 71.7 |
| **Age groups** | Up to 35 years old | 56 | 46.7 |
| | From 36 to 50 years old | 42 | 35.0 |
| | Over 50 years old | 22 | 18.3 |
| **Marital status** | Single | 30 | 25.0 |
| | Married/living with a partner | 86 | 71.7 |
| | Divorced | 2 | 1.7 |
| | Widowed | 2 | 1.7 |
| **Qualifications** | Bachelor | 4 | 3.3 |
| | Graduate | 74 | 61.7 |
| | Master's degree | 26 | 21.7 |
| | Doctorate | 16 | 13.3 |
| **Sector of activity** | Public sector (central government, local government, public administration) | 54 | 45.0 |
| | Private sector | 62 | 51.7 |
| | Social and non-profit sector (local NGO, youth group, religious organization, voluntary movement) | 2 | 1.7 |
| | International organization | 2 | 1.7 |
| **Nationality** | Angolan | 94 | 78.3 |
| | Portuguese | 22 | 18.3 |
| | Brazilian | 4 | 3.4 |

The vast majority of respondents are married/cohabiting (71.7%), with 30 single respondents (25%) and only two divorced and widowed respondents (1.7%), as can be seen in Table 3.

Concerning academic qualifications, as shown in the following table, the majority of the respondents have a Bachelor's degree (61.7%), 26 respondents have a Master's degree (21.7%), 16 respondents have a PhD (13.3%), and 4 respondents have a Bachelor's degree (3.3%).

Most of the respondents (see Table 3) work in the private sector (51.7%), followed very closely by those who work in the public sector (45%).

In terms of nationality (see Table 3), the majority of respondents are Angolan (78.3%), Portuguese (18.3%), and Brazilian (3.4%).

### 4.1. Analysis of the Results Obtained

A univariate data analysis was first performed using descriptive statistics (absolute and relative frequencies, mean, and standard deviation) for data analysis. Then, the internal consistency of the research instrument was assessed through Cronbach's alpha values. In the analysis of the reliability for all items making up the scale used, a Cronbach's alpha value of 0.882 (good internal consistency) was obtained, which could be considered to be a good result [30].

### 4.1.1. Evaluation of Strategic Goals for Sustainable Development and Competitiveness

To assess the strategic goals considered to be a priority for development and competitiveness, 14 items were used [38], with scores above the arithmetic mean in all the alternatives presented (see Table 4). As can be seen in this table, respondents considered diversification of the economy to be the most important, with 83 total concordances (M = 4.59, SD = 0.670), followed by the training of human resources, with 72 total concordances (M = 4.58, SD = 0.561); the development of transport and communications with 70 total concordances (M = 4.58, SD = 0.496); and the promotion of science, technology, innovation, and entrepreneurship with 66 total concordances (M = 4.46, SD = 0.675). The lowest values were obtained on the items "integrate unproductive or difficult to employ workers (M = 3.03, SD = 1.132), "invest in business parks and facilities" (M = 3.63, SD = 0.959), and "establish an adequate system of environmental protection, natural disaster prevention, and air and maritime safety" (M = 3.69, SD = 1.034).

**Table 4.** Priority strategic goals for sustainable development and competitiveness.

| Cronbach's Alpha 0.745 | Totally Disagree | | Disagree | | Neither Agree Nor Disagree | | Agree | | Totally Agree | | M | SD |
|---|---|---|---|---|---|---|---|---|---|---|---|---|
| | F | % | F | % | F | % | F | % | F | % | | |
| Improving the local business investment environment | | | | | 8 | 6.7 | 54 | 45.0 | 56 | 46.7 | 4.41 | 0.617 |
| Investing in tangible strategic infrastructure | | | | | 6 | 5.0 | 68 | 56.7 | 46 | 38.3 | 4.32 | 0.568 |
| Investing in business parks and facilities | 2 | 1.7 | 14 | 11.7 | 30 | 25.0 | 52 | 43.3 | 20 | 16.7 | 3.63 | 0.959 |
| Investing in intangible strategic infrastructure | 12 | 10.0 | 34 | 28.3 | 48 | 40.0 | 24 | 20.0 | | | 3.71 | 0.907 |
| Promoting local business growth | | | 6 | 5.0 | 6 | 5.0 | 50 | 41.7 | 58 | 48.3 | 4.32 | 0.794 |
| Promoting the creation of new companies | 2 | 1.7 | 8 | 6.7 | 22 | 18.3 | 52 | 43.3 | 36 | 30.0 | 3.92 | 0.948 |

**Table 4.** *Cont.*

| Cronbach's Alpha 0.745 | Totally Disagree | | Disagree | | Neither Agree Nor Disagree | | Agree | | Totally Agree | | M | SD |
|---|---|---|---|---|---|---|---|---|---|---|---|---|
| | **F** | **%** | **F** | **%** | **F** | **%** | **F** | **%** | **F** | **%** | | |
| Attracting foreign investment | | | 2 | 1.7 | 10 | 8.3 | 60 | 50.0 | 46 | 38.3 | 4.27 | 0.688 |
| Developing business sectors and clusters | | | 10 | 8.3 | 20 | 16.7 | 74 | 61.7 | 14 | 11.7 | 3.78 | 0.764 |
| Integrating unproductive or hard-to-employ workers | 16 | 13.3 | 18 | 15.0 | 40 | 33.3 | 36 | 30.0 | 8 | 6.7 | 3.02 | 1.132 |
| Establishing an adequate system of environmental protection, natural disaster prevention, and air and maritime safety | 4 | 3.3 | 12 | 10.0 | 26 | 21.7 | 50 | 41.7 | 28 | 23.3 | 3.69 | 1.034 |
| Develop, train, and empower human resources | | | | | 4 | 3.3 | 42 | 35.0 | 72 | 60.0 | 4.58 | 0.561 |
| Develop transportation and communications | | | | | | | 50 | 41.7 | 70 | 58.3 | 4.58 | 0.496 |
| Promote science, technology, innovation, and entrepreneurship | | | 2 | 1.7 | 6 | 5.0 | 46 | 38.3 | 66 | 55.0 | 4.46 | 0.675 |
| Diversify the economy | | | 2 | 1.7 | 6 | 5.0 | 30 | 25.0 | 82 | 68.3 | 4.59 | 0.670 |
| KMO test and Bartlett's test | Kaiser–Meyer–Olkin (KMO) of sampling adequacy | | | | | | | | | | 0.527 | |
| | Bartlett's test of sphericity | | | | | | Chi-square gl Sig. | | | | 508.056 91 0.000 | |

The reliability analysis for all items that make up the "priority goals" scale obtained a Cronbach's alpha of 0.745, which is considered to be a scale with good internal consistency [30]. In addition, the Kaiser–Meyer–Olkin (KMO) measure of sampling adequacy was positive and significant (KMO = 0.527, Bartlett's with sig = 0.000), causing no problems with data analysis, revealing positive correlations among the variables.

4.1.2. Analysis of Angola's Key Economic, Social, and Environmental Vulnerabilities

The assessment of Angola's main economic, social, and environmental vulnerabilities was based on previous works by the [38,39]. As shown in the Table 5, the majority of respondents (55%) consider that the main vulnerabilities are "corruption and informality of the economy" (M = 4.37, SD = 0.865), followed by bureaucracy (M = 4.27, SD = 0.940); oil dependency M = 4.27, SD = 0.993); and high costs of energy, infrastructure, transportation, and maintenance (M = 4.24, SD = 0.967). The results also show that the least valued economic, social, and environmental vulnerabilities were "fragile environments and vulnerabilities to natural disasters" (M = 3.02, SD = 1.132) and "a narrow resource base and little or no opportunity to create economies of scale" (M = 3.27, SD = 0.993). The internal reliability obtained on this dimension was good (Cronbach's Alpha = 0.899), and positive and significant results were obtained on the Kaiser–Meyer–Olkin (KMO) measure and Bartlett's test of sphericity (KMO = 0.527, Bartlett's sig = 0.000).

**Table 5.** Major economic, social, and environmental vulnerabilities in Angola.

| Cronbach's Alpha 0.899 | Totally Disagree | | Disagree | | Neither Agree Nor Disagree | | Agree | | Totally Agree | | M | SD |
|---|---|---|---|---|---|---|---|---|---|---|---|---|
| | F | % | F | % | F | % | F | % | F | % | | |
| Narrow resource base and little or no opportunity to create economies of scale | 10 | 8.3 | 18 | 15.0 | 30 | 25.0 | 58 | 48.3 | 4 | 3.3 | 3.27 | 0.993 |
| Small domestic markets, strong dependency on some foreign markets, distances to export, and resource import markets | 8 | 6.7 | 6 | 5.0 | 2 | 1.7 | 54 | 45.0 | 50 | 41.7 | 4.15 | 1.043 |
| High energy, infrastructure, transport, communication, and maintenance costs | 4 | 3.3 | 4 | 3.3 | 8 | 6.7 | 46 | 38.3 | 56 | 46.7 | 4.24 | 0.967 |
| Low and irregular international traffic volumes | 2 | 1.7 | 10 | 8.3 | 40 | 33.3 | 42 | 35.0 | 26 | 21.7 | 3.68 | 0.969 |
| Fragile natural environments and vulnerability to natural disasters | 12 | 10, | 34 | 28.3 | 22 | 18.3 | 46 | 38.3 | 6 | 5.0 | 3.02 | 1.132 |
| Small but growing population | 8 | 6.7 | 20 | 16.7 | 28 | 23.3 | 60 | 50.0 | 4 | 3.3 | 3.31 | 0.965 |
| High economic growth volatility | 2 | 1.7 | 22 | 18.3 | 36 | 30.0 | 52 | 43.3 | 8 | 6.7 | 3.37 | 0.904 |
| Limited opportunities for the private sector | 6 | 5.0 | 18 | 15.0 | 10 | 8.3 | 58 | 48.3 | 28 | 23.3 | 3.75 | 1.088 |
| A proportionately large dependency of the economy on its public sector | 4 | 3.3 | 6 | 5.0 | 12 | 10.0 | 60 | 50.0 | 38 | 31.7 | 4.07 | 0.884 |
| A disproportionately costly public administration | 2 | 1.7 | 10 | 8.3 | 14 | 11.7 | 52 | 43.3 | 42 | 35.0 | 4.07 | 0.903 |
| Corruption and informality of the economy | 4 | 3.3 | 2 | 1.7 | 12 | 10.0 | 36 | 30.0 | 66 | 55.0 | 4.37 | 0.865 |
| Restricted access to credit | 2 | 1.7 | 10 | 8.3 | 10 | 8.3 | 46 | 38.3 | 52 | 43.3 | 4.19 | 0.915 |
| Weak energy distribution systems | 4 | 3.3 | 8 | 6.7 | 10 | 8.3 | 58 | 48.3 | 40 | 33.3 | 4.07 | 0.922 |
| High inflation | 4 | 3.3 | 10 | 8.3 | 6 | 5.0 | 42 | 35.0 | 58 | 48.3 | 4.22 | 0.997 |
| Excessive bureaucracy | 6 | 5.0 | 4 | 3.3 | 4 | 3.3 | 50 | 41.7 | 56 | 46.7 | 4.27 | 0.940 |
| Inefficient judicial system | 2 | 1.7 | 12 | 10.0 | 14 | 11.7 | 48 | 40.0 | 44 | 36.7 | 4.05 | 0.950 |
| Unskilled labor force | 4 | 3.3 | 16 | 13.3 | 22 | 18.3 | 44 | 36.7 | 34 | 28.3 | 3.78 | 1.063 |
| Dependency on oil | 2 | 1.7 | 12 | 10.0 | 4 | 3.3 | 40 | 33.3 | 62 | 51.7 | 4.27 | 0.993 |
| Weak currency | 4 | 3.3 | 8 | 6.7 | 8 | 6.7 | 48 | 40.0 | 52 | 43.3 | 4.19 | 0.951 |
| Literacy/education of the pop. | 4 | 3.3 | 12 | 10.0 | 10 | 8.3 | 54 | 45.0 | 40 | 33.3 | 3.98 | 1.038 |

| KMO test and Bartlett's test | Kaiser–Meyer–Olkin (KMO) of sampling adequacy | 0.527 |
|---|---|---|
| | Bartlett's test of sphericity | Chi-square 1651.767<br>gl 190<br>Sig. 0.000 |

### 4.1.3. Impact of Clusters on the Development of the Economy and Business Competitiveness

The specialists questioned generally consider that the creation and development of clusters (for example, energy, agriculture, and livestock, etc.) influence the development of the Angolan economy [40] and promote the competitiveness of businesses and locations (see Table 6). Thus, the main impacts recognized are related to the promotion of business productivity (M = 4.07, SD = 0.560), the implementation of innovations (M = 4.04, SD = 0.797), and the formation of new businesses (M = 4.02, SD = 0.809).

**Table 6.** Influence of clusters on the development of the economy and business competitiveness.

| Cronbach's Alpha 0.816 | Totally Disagree | | Disagree | | Neither Agree Nor Disagree | | Agree | | Totally Agree | | M | SD |
|---|---|---|---|---|---|---|---|---|---|---|---|---|
| | F | % | F | % | F | % | F | % | F | % | | |
| Promote business and local competitiveness | 2 | 1.7 | 6 | 5.0 | 8 | 6.7 | 76 | 63.3 | 24 | 20.0 | 3.96 | 0.797 |
| Promote an increase in company productivity | | | 4 | 3.3 | 2 | 1.7 | 90 | 75.0 | 20 | 16.7 | 4.07 | 0.560 |
| Facilitate complementarities between the activities of the different actors | | | 4 | 3.3 | 28 | 23.3 | 64 | 53.3 | 20 | 16.7 | 3.84 | 0.724 |
| Facilitate access to institutions and benefits | | | 4 | 3.3 | 30 | 25.0 | 70 | 58.3 | 12 | 10.0 | 3.75 | 0.659 |
| Help measure the performance of domestic activities and limit opportunistic behavior | | | 14 | 11.7 | 46 | 38.3 | 44 | 36.7 | 10 | 8.3 | 3.44 | 0.820 |
| Facilitate the implementation of innovations | | | 8 | 6.7 | 10 | 8.3 | 66 | 55.0 | 30 | 25.0 | 4.04 | 0.797 |
| Facilitate the formation of new enterprises | | | 8 | 6.7 | 12 | 10.0 | 64 | 53.3 | 30 | 25.0 | 4.02 | 0.809 |
| KMO test and Bartlett's test | Kaiser–Meyer–Olkin (KMO) of sampling adequacy. | | | | | | | | | | 0.745 | |
| | Bartlett's test of sphericity | | | | | | Chi-square gl Sig. | | | | 277.846 21 0.000 | |

The items making up the scale obtained good reliability (alpha = 0.816), as did the Kaiser–Meyer–Olkin measure of sampling adequacy (KMO = 0.745) with a significant Bartlett's sphericity ($p$ = 0.000).

### 4.1.4. Assessment of Priority Goals for Angola's Economic Development

Concerning the priority goals for Angola's economic development (see Table 7), respondents consider that they emphasize the building of human capital through investments in education and training (M = 3.92, SD = 0.958) and address solutions to Angola's most critical issues, such as high levels of indebtedness, inadequate access to technology, difficulties with commercial transactions, and inadequate access to sources of financing (M = 3.55, SD = 0.887).

**Table 7.** Assessment of priority targets for Angola's economic development.

| Cronbach's Alpha 0.782 | Totally Disagree | | Disagree | | Neither Agree Nor Disagree | | Agree | | Totally Agree | | M | SD |
|---|---|---|---|---|---|---|---|---|---|---|---|---|
| | F | % | F | % | F | % | F | % | F | % | | |
| Give sufficient focus on innovation, creativity, and technological systems as part of a sustainable development strategy | | | 30 | 25.0 | 32 | 26.7 | 50 | 41.7 | 8 | 6.7 | 3.30 | 0.922 |
| Emphasis the building of human capital through investments in education and training | | | 16 | 13.3 | 12 | 10.0 | 58 | 48.3 | 34 | 28.3 | 3.92 | 0.958 |
| Give sufficient attention to the development or adoption of climate change resilience systems | 6 | 5.0 | 50 | 41.7 | 42 | 35.0 | 22 | 18.3 | | | 2.67 | 0.833 |
| Address Angola's most critical issues such as high debt levels, inadequate access to technology, difficulties with commercial transactions, and inadequate access to sources of finance | 2 | 1.7 | 16 | 13.3 | 26 | 21.7 | 66 | 55.0 | 10 | 8.3 | 3.55 | 0.887 |
| Present energy alternatives to break the dependency on oil, namely at the level of biofuels | 8 | 6.7 | 24 | 20.0 | 30 | 25.0 | 48 | 40.0 | 10 | 8.3 | 3.23 | 1.075 |
| Increase the rate of dependency and imbalances in all factors | | | 22 | 18.3 | 60 | 50.0 | 30 | 25.0 | 8 | 6.7 | 3.20 | 0.816 |
| KMO test and Bartlett's test | Kaiser–Meyer–Olkin (KMO) of sampling adequacy | | | | | | | | | | 0.812 | |
| | Bartlett's test of sphericity | | | | | | Chi-square gl Sig. | | | | 170.754 15 0.000 | |

The least valued item was "pay sufficient attention to the development or adoption of climate resilience systems" (M = 2.67, SD = 0.833), which may mean that policymakers (and even Angolan citizens) are more concerned with short-term measures to mitigate current problems, to reduce the economic and social vulnerability of populations, and to protect their livelihoods. The reliability analysis for all items that make up the scale used obtain a Cronbach's alpha value of 0.782, which can be considered to be a good result (Pestana & Gageiro, 2005). Similarly, the KMO and Bartlett's test show high and significant values (KMO = 0.812, Bartlett's with sig = 0.000), showing that the variables are related.

4.1.5. Time Horizon for the Development of an Alternative Diversification Model

The period of time that respondents consider acceptable/realistic to develop an alternative model for a diversification strategy for Angola's economic development (see Table 8) is 10–20 years (48.3%). However, 36 respondents (30%) consider that developing an alternative model will take between 5 and 10 years, 16 specialists (13.3%) consider that between 20 and 30 years will be necessary, and 10 specialists (8.3%) consider that between 30 and 40 years will be necessary.

**Table 8.** Time frame for the development of an alternative diversification model.

| Time Period | F | % |
|---|---|---|
| 5–10 years | 36 | 30.0 |
| 10–20 years | 58 | 48.3 |
| 20–30 years | 16 | 13.3 |
| 30–40 years | 10 | 8.3 |

### 4.1.6. Priority Sectors to Be Developed in Angola

The priority sectors (Francisco Miguel Paulo—CEIC) that the respondents consider a priority to develop in Angola (see Table 9) are agriculture, livestock, and forestry (74); manufacturing (16); and tourism (10). Oil and gas (2), trade and distribution, and construction appear in an intermediate position. Diamonds and precious stones and services are in the last positions.

**Table 9.** Priority activity sectors to be developed in Angola.

| | Sorting Priority | | Mean |
|---|---|---|---|
| | F | % | |
| Agriculture, livestock, and forestry | 74 | 61.7 | 1.52 |
| Manufacturing industry | 16 | 13.3 | 3.04 |
| Tourism | 10 | 8.3 | 4.39 |
| Oil and gas | 2 | 1.7 | 5.00 |
| Trade and distribution | | | 5.34 |
| Building and construction | 2 | 1.7 | 5.70 |
| Diamonds and precious stones | | | 6.11 |
| Services | 2 | 1.7 | 6.23 |

### 4.1.7. Government Measures to Diversify the Angolan Economy

To assess government measures for diversifying the Angolan economy, the work of [39] was considered. In the reliability analysis for all items, a Cronbach's alpha value of 0.728 was obtained, thus, revealing good internal consistency. Bartlett's test of sphericity evidences that the variables are correlated in the population (KMO = 0.670, Bartlett's sig = 0.000). As shown in the Table 10, respondents consider that the main measures are technological innovation (M = 4.47, SD = 0.593), actions to enhance human capital (M = 4.38, SD = 0.780), tax regimes and financial incentive systems friendly to diversification and stimulating private investment (M = 4.38, SD = 0.663), and economic policies (M = 4.37, SD = 0.634). The least valued, although positive, measures were export subsidies (M = 3.60, SD = 0.956) and tax cuts (M = 3.67, SD = 0.892).

**Table 10.** Measures that the government should take to diversify the Angolan economy.

| Cronbach's Alpha 0.728 | Totally Disagree | | Disagree | | Neither Agree Nor Disagree | | Agree | | Totally Agree | | M | DP |
|---|---|---|---|---|---|---|---|---|---|---|---|---|
| | F | % | F | % | F | % | F | % | F | % | | |
| Tax reduction | | | 14 | 11.7 | 32 | 26.7 | 54 | 45.0 | 20 | 16.7 | 3.67 | 0.892 |
| Subsidised interest rates | | | 12 | 10.0 | 26 | 21.7 | 56 | 46.7 | 26 | 21.7 | 3.80 | 0.894 |
| Economic policies | | | | | 10 | 8.3 | 56 | 46.7 | 54 | 45.0 | 4.37 | 0.634 |
| Human capital development measures | 4 | 3.3 | | | 10 | 8.3 | 42 | 35.0 | 64 | 53.3 | 4.38 | 0.780 |
| Export subsidies | | | 18 | 15.0 | 34 | 28.3 | 46 | 38.3 | 22 | 18.3 | 3.60 | 0.956 |

**Table 10.** *Cont.*

| Cronbach's Alpha 0.728 | Totally Disagree | | Disagree | | Neither Agree Nor Disagree | | Agree | | Totally Agree | | M | DP |
|---|---|---|---|---|---|---|---|---|---|---|---|---|
| | F | % | F | % | F | % | F | % | F | % | | |
| Creation of agendas for diversification and national agencies with the responsibility of stimulating and coordinating the process of structural changes | 2 | 1.7 | 4 | 3.3 | 20 | 16.7 | 72 | 60.0 | 22 | 18.3 | 3.90 | 0.793 |
| Organizing meetings, lectures, seminars, workshops, etc., to inculcate a new spirit of greater openness to international competition | | | 6 | 5.0 | 18 | 15.0 | 80 | 66.7 | 16 | 13.3 | 3.88 | 0.688 |
| Technological innovation | | | | | 6 | 5.0 | 52 | 43.3 | 62 | 51.7 | 4.47 | 0.593 |
| Tax regimes and systems of financial incentives that are friendly to diversification and stimulate private investment | | | | | 12 | 10.0 | 50 | 41.7 | 58 | 48.3 | 4.38 | 0.663 |

| KMO test and Bartlett's test | Kaiser–Meyer–Olkin (KMO) of sampling adequacy | | 0.670 |
|---|---|---|---|
| | Bartlett's test of sphericity | Chi-square gl Sig. | 271.550 36 0.000 |

### 4.1.8. Strategic Alternatives for Diversifying the Economy

In order to evaluate the strategic alternatives for diversification of the Angolan economy, based on the works identified in the literature [39,44], and as presented in the Table 11, the specialists surveyed considered the most relevant strategies to be "harnessing of national raw materials", with 58.3% of total concordances (M = 4.55, SD = 0.563), "valorization of national human resources", with 60% of total concordances (M = 4.45, SD = 0.787), and "valorization of human potential and innovation", with 51.7% of concordances (M = 4.42, SD = 0.693).

**Table 11.** Strategic alternatives for diversifying the economy.

| Cronbach's Alpha 0.801 | Totally Disagree | | Disagree | | Neither Agree Nor Disagree | | Agree | | Totally Agree | | M | SD |
|---|---|---|---|---|---|---|---|---|---|---|---|---|
| | F | % | F | % | F | % | F | % | F | % | | |
| Import substitution (through efficiency and not through administrative protection mechanisms, which only generate bureaucracy and corruption), | 2 | 1.7 | 6 | 5.0 | 12 | 10.0 | 62 | 51.7 | 38 | 31.7 | 4.07 | 0.877 |
| Production of intermediate products | | | 2 | 1.7 | 12 | 10.0 | 78 | 65.0 | 28 | 23.3 | 4.10 | 0.627 |

**Table 11.** *Cont.*

| Cronbach's Alpha 0.801 | Totally Disagree | | Disagree | | Neither Agree Nor Disagree | | Agree | | Totally Agree | | M | SD |
|---|---|---|---|---|---|---|---|---|---|---|---|---|
| | F | % | F | % | F | % | F | % | F | % | | |
| Valorization of national human resources (reducing dependence on expatriates), | | | 4 | 3.3 | 10 | 8.3 | 34 | 28.3 | 72 | 60.0 | 4.45 | 0.787 |
| Technological innovation | | | 6 | 5.0 | 2 | 1.7 | 58 | 48.3 | 54 | 45.0 | 4.33 | 0.748 |
| Use of national raw materials | | | | | 4 | 3.3 | 46 | 38.3 | 70 | 58.3 | 4.55 | 0.563 |
| Diversify the destination of exports. | 2 | 1.7 | 2 | 1.7 | 18 | 15.0 | 56 | 46.7 | 42 | 35.0 | 4.12 | 0.842 |
| Definition of long-term industrial and investment policies | | | 2 | 1.7 | 20 | 16.7 | 68 | 56.7 | 30 | 25.0 | 4.05 | 0.696 |
| Accentuation of the role of financing the national investment system and the development of the financial and banking sector | | | 2 | 1.7 | 26 | 21.7 | 62 | 51.7 | 30 | 25.0 | 4.00 | 0.733 |
| The valorization of human potential and innovation | | | 2 | 1.7 | 8 | 6.7 | 48 | 40.0 | 62 | 51.7 | 4.42 | 0.693 |
| The emergence of specializations around the reinforcement of supply based on new producers and new products | | | 2 | 1.7 | 18 | 15.0 | 74 | 61.7 | 26 | 21.7 | 4.03 | 0.660 |

| KMO test and Bartlett's test | Kaiser-Meyer-Olkin (KMO) of sampling adequacy | | 0.764 |
|---|---|---|---|
| | Bartlett's test of sphericity | Chi-square<br>gl<br>Sig. | 465.523<br>55<br>0.000 |

Reliability analysis on the totality of the items revealed a good internal consistency ($\alpha$ = 0.801), and the Kaiser–Meyer–Olkin measure and sampling adequacy (KMO = 0.764) revealed positive and significant correlations among the variables (Bartlett with sig = 0.000).

4.1.9. Reasons for a Lack of Competitiveness in Angola's Oil Sector

Respondents point to the lack of competitiveness in Angola's oil sector, based on the Economist Intelligence Unit (EIU), as being mainly due to "the complicated web of financial arrangements created by oil-backed loans" (M = 3.97, SD = 0.777) and "the inaccuracy of government expenditure declarations" (M = 3.92, SD = 0.922) (Table 12).

**Table 12.** Reasons for the lack of competitiveness of Angola's oil sector.

| Cronbach's Alpha 0.825 | Totally Disagree | | Disagree | | Neither Agree Nor Disagree | | Agree | | Totally Agree | | M | SD |
|---|---|---|---|---|---|---|---|---|---|---|---|---|
| | F | % | F | % | F | % | F | % | F | % | | |
| Revenues Sonangol receives from taxes, joint ventures, and other sources of income do not appear in government accounts | 4 | 3.3 | 18 | 15.0 | 46 | 38.3 | 38 | 31.7 | 14 | 11.7 | 3.33 | 0.982 |
| Oil prices are undervalued in the government budget, and any revenue above this estimate is never declared | 6 | 5.0 | 10 | 8.3 | 20 | 16.7 | 54 | 45.0 | 30 | 25.0 | 3.77 | 1.075 |
| Government expenditure declarations are not accurate | 4 | 3.3 | 4 | 3.3 | 20 | 16.7 | 62 | 51.7 | 30 | 25.0 | 3.92 | 0.922 |
| The share of taxes and royalties that Sonangol actually pays to the government is transferred with significant delay and in local currency | | | 10 | 8.3 | 48 | 40.0 | 36 | 30.0 | 26 | 21.7 | 3.65 | 0.913 |
| The web of financial arrangements created by oil-backed loans is complicated | | | 4 | 3.3 | 26 | 21.7 | 60 | 50 | 30 | 25.0 | 3.97 | 0.777 |

| KMO test and Bartlett's test | Kaiser–Meyer–Olkin (KMO) of sampling adequacy | | 0.732 |
|---|---|---|---|
| | Bartlett's test of sphericity | Chi-square | 237.865 |
| | | gl | 10 |
| | | Sig. | 0.000 |

The reliability analysis through Cronbach's alpha revealed good internal consistency ($\alpha = 0.825$), also verifying that there are correlations among the different items (KMO = 0.732, Bartlett's with sig = 0.000).

4.1.10. Energy Alternatives

When asked to evaluate the best energy alternatives to reduce dependency on oil, most of the respondents (see Table 13) put solar energy as the priority (48), followed by biodiesel (22), hydraulic energy (14), and bioethanol (8).

**Table 13.** The best alternatives to oil to be explored in Angola.

| Energy Alternatives | Ranking of Alternatives |
|---|---|
| Solar | 1st priority (48) |
| Biodiesel | 2nd priority (22) |
| Hydro | 3rd priority (14) |
| Bioethanol | 4th priority (8) |
| Wind | 5th priority (2) |
| Geothermal | Priority 6th (2) |
| Offshore | Priority 7th (2) |
| Biodiesel from algae | 8th priority (2) |
| H-BIO | 9th priority (0) |

The least valued alternatives were H-bio, biodiesel from algae, and geothermal and wind energy.

### 4.1.11. Assessing the Attractiveness of Biofuels

In order to assess the attractiveness of biofuels, we considered works developed by [50,51] and, in particular, four alternatives: bioethanol, biodiesel, H-bio, and algae. The analysis of Cronbach's alpha allows us to conclude that there is good reliability of the scale ($\alpha = 0.711$), and the Kaiser–Meyer–Olkin measure of sampling adequacy attests to the correlations among the variables (KMO = 0.615, Bartlett with sig = 0.000). Thus, regarding bioethanol, in general, respondents rated this biofuel as attractive (M = 3.274), having obtained a value below 3.5 (arithmetic mean) in most criteria (Table 14).

**Table 14.** Assessing the attractiveness of biofuels.

| Cronbach Alpha 0.711 | Extreme | | Very Strong | | Strong | | Moderate | | Weak | | Very Weak | | Nil | | M | SD |
|---|---|---|---|---|---|---|---|---|---|---|---|---|---|---|---|---|
| | F | % | F | % | F | % | F | % | F | % | F | % | F | % | | |
| Criterion 1: Cost of technology | 8 | 6.7 | 4 | 3.3 | 26 | 21.7 | 28 | 23.3 | 6 | 5.0 | 4 | 3.3 | 76 | 63.3 | 3.65 | 1.033 |
| Criterion 2: Emission of pollutant gases | 4 | 3.3 | 22 | 18.3 | 16 | 13.3 | 8 | 6.7 | 12 | 10.0 | 4 | 3.3 | 4 | 3.3 | 3.35 | 1.581 |
| Criterion 3: Employment generation | 14 | 11.7 | 18 | 15.0 | 26 | 21.7 | 14 | 11.7 | 2 | 1.7 | 4 | 3.3 | | | 2.82 | 1.281 |
| Criterion 4: Productivity | 10 | 8.3 | 14 | 11.7 | 32 | 26.7 | 6 | 5.0 | 4 | 3.3 | 6 | 5.0 | 2 | 1.7 | 3.18 | 1.455 |
| Criterion 5: Natural resources | 10 | 8.3 | 20 | 16.7 | 22 | 18.3 | 10 | 8.3 | 10 | 8.3 | 2 | 1.7 | | | 2.88 | 1.264 |
| Criterion 6: Contribution to competitiveness | 14 | 11.7 | 8 | 6.7 | 14 | 11.7 | 30 | 25.0 | | | 4 | 3.3 | 2 | 1.7 | 3.26 | 1.512 |
| Criterion 7: Energy potential | 16 | 13.3 | 10 | 8.3 | 10 | 8.3 | 22 | 18.3 | 8 | 6.7 | 2 | 1.7 | 4 | 3.3 | 3.35 | 1.691 |
| Criterion 8: Innovation and change | 10 | 8.3 | 12 | 10.0 | 10 | 8.3 | 20 | 16.7 | 12 | 10.0 | 2 | 1.7 | 4 | 3.3 | 3.53 | 1.643 |
| KMO test and Bartlett's test | Kaiser-Meyer-Olkin (KMO) of sampling adequacy | | | | | | | | | | | | | | 0.615 | |
| | Bartlett's test of sphericity | | | | | | | | | | | Chi-square gl Sig. | | | 161.223 28 0.000 | |

As can be seen in the Table 14, the most valued criteria were Criterion 3 (generation of jobs) (11.7% of respondents rated it as "strong", 15% as "very strong", and 21.7% as "strong" (M = 2.82, SD = 1.281)), and Criterion 5 (existence of natural resources) (rated as "strong" by 18.3% of respondents, as "very strong" by 16.7% of respondents, and as "extreme" by 8.3% of respondents (M = 2.88, SD = 1.264)). However, two criteria were rated slightly higher than the arithmetic mean (3.5): Criterion 1 (cost of technology developed for production (M = 3.65, SD = 1.033)) and Criterion 8 (innovation and systemic change (M = 3.53, SD = 1.643)), and therefore, were considered to be the least attractive factors for this biofuel by the experts surveyed.

Regarding biodiesel, the respondents evaluate this biofuel as attractive (M = 3.356) (Table 15).

The reliability analysis on the totality of the items revealed good internal consistency ($\alpha = 0.747$). Similarly, significant correlations were found among the items that make up the scale (KMO = 0.550, Bartlett's with sig = 0.000). As presented in the Table 15, the best-ranked criteria were Criterion 3 (generation of jobs), which was rated as "very strong" by 20% of the respondents, as "strong" by 15% of the respondents, and as "extreme"

by 10% of the respondents (M = 3.00, SD = 1.509) and Criterion 7, (energy potential), which was rated as "very strong" by 20% of the respondents and as "extreme" by 10% of the respondents (M = 3.18, SD = 1.578). Criteria 2 (gas emissions) and 5 (existence of natural resources) were similar in terms of average (M = 3.24), Criterion 1 (cost of technology developed for production (M = 3.70; SD = 1.176)), Criterion 6 (contribution to the country's competitiveness (M = 3.70, SD = 1.617)), and Criterion 8 (innovation and systemic change (M = 3.52; SD = 1.491)) were rated as less attractive with a result higher than the arithmetic mean.

**Table 15.** Biodiesel attractiveness assessment.

| Cronbach's Alpha 0.747 | Extreme | | Very Strong | | Strong | | Moderate | | Weak | | Very Weak | | Nil | | M | SD |
|---|---|---|---|---|---|---|---|---|---|---|---|---|---|---|---|---|
| | F | % | F | % | F | % | F | % | F | % | F | % | F | % | | |
| Criterion 1: Cost of technology | 4 | 3.3 | 8 | 6.7 | 14 | 11.7 | 28 | 23.3 | 14 | 11.7 | 2 | 1.7 | | | 3.70 | 1.176 |
| Criterion 2: Emission of pollutant gases | 8 | 6.7 | 28 | 23.3 | 6 | 5.0 | 10 | 8.3 | 14 | 11.7 | 4 | 3.3 | 4 | 3.3 | 3.24 | 1.665 |
| Criterion 3: Employment generation | 12 | 10.0 | 24 | 20.0 | 18 | 15.0 | 14 | 11.7 | 2 | 1.7 | 4 | 3.3 | 2 | 1.7 | 3.00 | 1.509 |
| Criterion 4: Productivity | 6 | 5.0 | 14 | 11.7 | 26 | 21.7 | 10 | 8.3 | 8 | 6.7 | 6 | 5.0 | | | 3.27 | 1.387 |
| Criterion 5: Natural resources | 12 | 10.0 | 16 | 13.3 | 8 | 6.7 | 22 | 18.3 | 4 | 3.3 | 8 | 6.7 | | | 3.24 | 1.570 |
| Criterion 6: Contribution to competitiveness | 10 | 8.3 | 10 | 8.3 | 8 | 6.7 | 24 | 20.0 | 14 | 11.7 | 6 | 5.0 | 2 | 1.7 | 3.70 | 1.617 |
| Criterion 7: Energy potential | 12 | 10.0 | 24 | 20.0 | 8 | 6.7 | 16 | 13.3 | 14 | 11.7 | 2 | 1.7 | | | 3.18 | 1.578 |
| Criterion 8: Innovation and change | 8 | 6.7 | 12 | 10.0 | 10 | 8.3 | 32 | 26.7 | 6 | 5.0 | 4 | 3.3 | 2 | 1.7 | 3.52 | 1.491 |
| KMO test and Bartlett's test | Kaiser–Meyer–Olkin (KMO) of sampling adequacy | | | | | | | | | | | | | | 0.550 | |
| | Bartlett's test of sphericity | | | | | | | | | Chi-square gl Sig. | | | | | 236.730 28 0.000 | |

The evaluation of attractiveness of H-bio was assessed as uninteresting with a result higher than the arithmetic mean (M = 3.782) (Table 16).

The scale showed good internal consistency, with a Cronbach's alpha value of 0.835. There were also correlations among items (KMO = 0.760, Bartlett test sig = 0.000), and only two criteria were considered to be attractive, i.e., Criterion 7 (energy potential), rated as "strong" by 16.7% of respondents, as "very strong" by 11.7% of respondents, and as "extreme" by 5% of respondents (M = 3.35, SD = 1.631) and Criterion 8 (innovation and systemic change), rated as "strong" by 18.3% of respondents, as "extreme" by 10% of respondents, and as "very strong" by 6.7% of respondents (M = 3.39, SD = 1.653). Among the remaining criteria with less attractive evaluations, Criterion 2 (emission of pollutant gases (M = 4.81, SD = 1.502)) and Criterion 6 (contribution to the country's competitiveness (M = 3.81, SD = 1.458)) stand out.

Finally, the assessment of the attractiveness of the biofuel from algae was considered to be unattractive or not attractive (M = 4.133), with all criteria scoring higher than the arithmetic mean. The worst score was obtained on Criterion 2 (emission of pollutant gases) (M = 4.78, SD = 1.777) and on Criterion 6 (contribution to the country's competitiveness (M = 4.28, SD = 1.804)). The best scores were obtained in Criterion 4 (productivity (M = 3.81, SD = 1.859)) and Criterion 5 (existence of natural resources (M = 3.91, SD = 1.892)). The

scale revealed good internal consistency, obtaining a Cronbach's alpha value of 0.798. The Kaiser–Meyer–Olkin (KMO) measure of sampling adequacy and Bartlett's test of sphericity showed that the variables were correlated in the population (KMO = 0.683, Bartlett's test sig. 0.000) (Table 17).

**Table 16.** Evaluation of the attractiveness of H-Bio.

| Cronbach's Alpha 0.835 | Extreme | | Very Strong | | Strong | | Moderate | | Weak | | Very Weak | | Nil | | M | SD |
|---|---|---|---|---|---|---|---|---|---|---|---|---|---|---|---|---|
| | F | % | F | % | F | % | F | % | F | % | F | % | F | % | | |
| Criterion 1: Cost of technology | 4 | 3.3 | 10 | 8.3 | 14 | 11.7 | 14 | 11.7 | 6 | 5.0 | 10 | 8.3 | 6 | 5.0 | 3.94 | 1.754 |
| Criterion 2: Emission of pollutant gases | 4 | 3.3 | 4 | 3.3 | 6 | 5.0 | 14 | 11.7 | 14 | 11.7 | 18 | 15.0 | 6 | 5.0 | 4.81 | 1.502 |
| Criterion 3: Employment generation | 4 | 3.3 | 2 | 1.7 | 22 | 18.3 | 24 | 20.0 | 4 | 3.3 | 6 | 5.0 | 2 | 1.7 | 3.74 | 1.330 |
| Criterion 4: Productivity | 8 | 6.7 | 8 | 6.7 | 20 | 16.7 | 16 | 13.3 | 4 | 3.3 | 6 | 5.0 | 6 | 5.0 | 3.58 | 1.713 |
| Criterion 5: Natural resources | 10 | 8.3 | 4 | 3.3 | 18 | 15.0 | 18 | 15.0 | 4 | 3.3 | 8 | 6.7 | 4 | 3.3 | 3.65 | 1.631 |
| Criterion 6: Contribution to competitiveness | 6 | 5.0 | 6 | 5.0 | 18 | 15.0 | 24 | 20.0 | 2 | 1.7 | 6 | 5.0 | 4 | 3.3 | 3.81 | 1.458 |
| Criterion 7: Energy potential | 6 | 5.0 | 14 | 11.7 | 20 | 16.7 | 8 | 6.7 | 8 | 6.7 | 4 | 3.3 | 4 | 3.3 | 3.35 | 1.631 |
| Criterion 8: Innovation and change | 12 | 10.0 | 8 | 6.7 | 22 | 18.3 | 10 | 8.3 | 10 | 8.3 | 2 | 1.7 | 6 | 5.0 | 3.39 | 1.653 |
| KMO test and Bartlett's test | Kaiser–Meyer–Olkin (KMO) of sampling adequacy | | | | | | | | | | | | | 0.760 | | |
| | Bartlett's test of sphericity | | | | | | | | | | Chi-square gl Sig. | | | | 279.155 28 0.000 | | |

**Table 17.** Algae attractiveness assessment.

| Cronbach's Alpha 0.798 | Extreme | | Very Strong | | Strong | | Moderate | | Weak | | Very Weak | | Nil | | M | SD |
|---|---|---|---|---|---|---|---|---|---|---|---|---|---|---|---|---|
| | F | % | F | % | F | % | F | % | F | % | F | % | F | % | | |
| Criterion 1: Cost of technology | 6 | 5.0 | 16 | 13.3 | 16 | 13.3 | 14 | 11.7 | 2 | 1.7 | 12 | 10.0 | 8 | 6.7 | 4.00 | 1.764 |
| Criterion 2: Emission of pollutant gases | 4 | 3.3 | 10 | 8.3 | 4 | 3.3 | 4 | 3.3 | 22 | 18.3 | 14 | 11.7 | 12 | 10.0 | 4.78 | 1.777 |
| Criterion 3: Employment generation | 6 | 5.0 | 8 | 6.7 | 20 | 16.7 | 14 | 11.7 | 6 | 5.0 | 8 | 6.7 | 8 | 6.7 | 3.97 | 1.699 |
| Criterion 4: Productivity | 10 | 8.3 | 10 | 8.3 | 14 | 11.7 | 10 | 8.3 | 10 | 8.3 | 10 | 8.3 | 6 | 5.0 | 3.81 | 1.859 |
| Criterion 5: Natural resources | 14 | 11.7 | 6 | 5.0 | 12 | 10.0 | 22 | 18.3 | 4 | 3.3 | 6 | 5.0 | 12 | 10.0 | 3.91 | 1.892 |
| Criterion 6: Contribution to competitiveness | 8 | 6.7 | 4 | 3.3 | 4 | 3.3 | 20 | 16.7 | 14 | 11.7 | 10 | 8.3 | 8 | 6.7 | 4.28 | 1.804 |

**Table 17.** *Cont.*

| Cronbach's Alpha 0.798 | Extreme | | Very Strong | | Strong | | Moderate | | Weak | | Very Weak | | Nil | | M | SD |
|---|---|---|---|---|---|---|---|---|---|---|---|---|---|---|---|---|
| | F | % | F | % | F | % | F | % | F | % | F | % | F | % | | |
| Criterion 7: Energy potential | 12 | 10.0 | 4 | 3.3 | 8 | 6.7 | 10 | 8.3 | 22 | 18.3 | 4 | 3.3 | 8 | 6.7 | 4.06 | 1.851 |
| Criterion 8: Innovation and change | 6 | 5.0 | 4 | 3.3 | 10 | 8.3 | 22 | 18.3 | 12 | 10.0 | 10 | 8.3 | 4 | 3.3 | 4.09 | 1.498 |

| KMO test and Bartlett's test | Kaiser–Meyer–Olkin (KMO) of sampling adequacy | | 0.683 |
|---|---|---|---|
| | Bartlett's test of sphericity | Chi-square gl Sig. | 204.586 28 0.000 |

In comparative terms, and as presented in the Table 18, the best alternatives are bioethanol (M = 3.254), followed by biodiesel (M = 3.356).

**Table 18.** Comparison between alternatives (average values).

| Criteria | Bioethanol M | Biodiesel M | H-BIO M | Algae M |
|---|---|---|---|---|
| Criterion 1: Cost of technology developed for production | 3.65 | 3.70 | 3.94 | 4.00 |
| Criterion 2: Emission of pollutant gases due to combustion in engine | 3.35 | 3.24 | 4.81 | 4.78 |
| Criterion 3: Job creation | 2.82 | 3.00 | 3.74 | 3.97 |
| Criterion 4: Productivity of raw material | 3.18 | 3.27 | 3.58 | 3.81 |
| Criterion 5: Existence of natural resources | 2.88 | 3.24 | 3.65 | 3.91 |
| Criterion 6: Contribution to country's competitiveness | 3.26 | 3.70 | 3.81 | 4.28 |
| Criterion 7: Energy potential | 3.35 | 3.18 | 3.35 | 4.06 |
| Criterion 8: Innovation and systemic change | 3.53 | 3.52 | 3.39 | 4.09 |
| TOTAL | 26.02 | 26.85 | 30.27 | 32.9 |
| Item mean | 3.254 | 3.356 | 3.782 | 4.113 |
| Correlation between items | 0.235 | 0.269 | 0.388 | 0.331 |

H-bio (M = 3.782) and biofuel from algae (M = 4.113) are considered to be unattractive (values higher than the average), the latter having the worst classification among all criteria (all higher than the arithmetic average).

## 5. Discussion of Results and Conclusions

The results obtained in the empirical research, in general, coincide with the observations, evaluations, suggestions, and recommendations presented by the professionals who participated in the focus group. Thus, the strategies considered to be priorities for development and competitiveness include diversification of the economy; training and education of human resources; development of transportation and communications; and promotion of science, technology, innovation, and entrepreneurship, as pointed out by the specialists and by the [38]. Angola's main economic, social, and environmental vulnerabilities are related to corruption and the informality of the economy; bureaucracy; dependency on oil; and the high costs of energy, infrastructure, transportation and maintenance, which are factors that have been already highlighted in the work of [39]. As suggested by [40], the specialists consider that the creation and development of regional clusters influence the development of Angola's economy and promote the competitiveness of companies and locals since they contribute to the productivity of companies and the implementation of

innovations and the formation of new companies. Regarding the priority goals for Angola's economic development, the professionals surveyed consider that they value the building of human capital through investments in education and training and contemplate solutions for Angola's most critical issues such as high levels of indebtedness, inadequate access to technology, difficulties with commercial transactions, and inadequate access to sources of financing. Most respondents consider it acceptable/realistic that between "10 and 20 years" will be required to develop an alternative model for Angola's diversification strategy for economic development. This time frame is aligned with the recommendations made by the professionals who participated in the focus group. However, this time horizon is shorter than the scenarios projected by international organizations [38] and studies identified in the literature [52]. The priority sectors that the respondents consider a priority to develop in Angola are agriculture, livestock and forestry, manufacturing, and tourism, corroborated by the specialists who participated in the focus group. The main government measures to diversify the Angolan economy target technological innovation, actions to enhance human capital, tax regimes, and financial incentive systems that are friendly to diversification and stimulate private investment and economic policies, similar to what was already pointed out in the work of [39]. The professionals consider that the strategic alternatives for diversification of the Angolan economy include the use of national raw materials (development of endogenous resources), the valorization of national human resources, and the valorization of human potential and innovation, as already considered in studies by [39,44]. As noted by the Economist Intelligence Unit (2018), respondents point out that the lack of competitiveness of Angola's oil sector is related to the "complicated web of financial arrangements created by oil-backed loans and the inaccuracy of government expenditure declarations. The best energy alternatives [48] to reduce oil dependence are, according to respondents, solar energy, biodiesel, hydropower, and bioethanol. In assessing the attractiveness and potential of biofuels [48,49], the best alternatives have been bioethanol, followed by biodiesel. In summary, the specialists and professionals who participated in this research consider it a priority, and an urgent priority, to diversify the Angolan economy based on a sustainable development model, supported by the valorization of endogenous resources (primary sector and industrialization of the country) through the promotion of education, a reduction in external dependency (potentialize the immense existing resources, i.e., internal production), the leveraging of exports, as well as a reduction in dependency on oil, through the exploitation of biofuels.

The results show that the strategies considered to be priorities for development and competitiveness include diversification of the economy; training and education of human resources; the development of transportation and communications; and the promotion of science, technology, innovation, and entrepreneurship. On the one hand, the professionals consulted consider that Angola's main economic, social, and environmental vulnerabilities are related to corruption and the informality of the economy; bureaucracy; dependency on oil and high energy; and infrastructure, transport, and maintenance costs. On the other hand, the research participants positively evaluate the creation and development of regional clusters, considering their influence on the development of Angola's economy and the promotion of business and territorial competitiveness. In line with the studies and works identified in the literature, the specialists consider that the priority goals for Angola's economic development must prioritize the valorization of human capital (investments in education, training, and capacity building) and that policies should be developed to solve the high levels of indebtedness, inadequate access to technology, difficulties with commercial transactions, and inadequate access to sources of financing. The sectors that respondents consider to be a priority to develop in Angola are agriculture, livestock and forestry, manufacturing, and tourism. The main government measures to diversify the Angolan economy should focus on technological innovation, actions to enhance human capital, tax regimes, and systems of financial incentives that are friendly to diversification and stimulate private investment and economic policies.

The professionals consider that the strategic alternatives for diversification of the Angolan economy include taking advantage of national raw materials (development of endogenous resources), valuing national human resources, and valuing human potential and innovation. The lack of competitiveness of the oil sector in Angola is related to the complicated network of financial arrangements created by loans guaranteed with oil and the inaccuracy of government spending declarations. According to the respondents, the best energy alternatives for reducing dependency on oil are solar energy, biodiesel, hydroelectric power, and bioethanol. When assessing the attractiveness and potential of biofuels, the best alternative is bioethanol, followed by biodiesel.

In summary, the specialists and professionals who participated in this research consider it a priority, and an urgent priority, to diversify the Angolan economy based on a sustainable development model, supported by the valorization of endogenous resources (primary sector and industrialization of the country) through the promotion of education, a reduction in external dependency (potentialize the immense existing resources, i.e., internal production), leveraging of exports, as well as a reduction in oil dependency through the exploitation of biofuels.

### 6. Limitations and Future Research

It is important to draw attention to two limitations associated with this research. One limitation is the methodological nature, more precisely the depth and scope of the research technique used. The other limitation is associated with the detailed analysis of the strategic alternative proposed by the theme for diversification of the Angolan economy.

Concerning the methodological limitation, the "Delphi" technique used for the research has advantages. It brings together specialists in the field of knowledge who can present precise and coherent points with scientific support, experience, and research on the subject matter. However, at the same time, their opinions, despite being valid and well supported, always carry some subjectivity and/or personal bias that must be compared, analyzed, and proven.

**Author Contributions:** Conceptualization, A.C., G.M., M.M. and R.S.; methodology, R.S.; software, M.M.; validation, R.S., G.M., A.C., M.M. and R.R.; formal analysis, R.S. and R.R.; investigation, G.M.; resources, R.R. and A.C.; data curation, R.S.; writing—original draft preparation, G.M.; writing—review and editing, M.M.; visualization, R.R., A.C., G.M., M.M. and R.S.; supervision, G.M.; project administration, G.M., M.M. and R.S.; funding acquisition, G.M., A.C., M.M., R.R. and R.S. All authors have read and agreed to the published version of the manuscript.

**Funding:** The work of author Rui Silva is supported by national funds, through the Portuguese Foundation for Science and Technology (FCT) under project UIDB/04011/2022 and by NECE-UBI, Research Centre for Business Sciences, Research Centre under project UIDB/04630/2022. The work of author Carmem Leal is supported by national funds, through the Portuguese Foundation for Science and Technology (FCT) under project UIDB/04011/2022.

**Institutional Review Board Statement:** The study was conducted in accordance with the Declaration of Helsinki and approved by the Institutional Review Board.

**Informed Consent Statement:** Informed consent was obtained from all subjects involved in the study.

**Data Availability Statement:** Not applicable.

**Acknowledgments:** The authors gratefully acknowledge the University of Trás-os-Montes and Alto Douro, CETRAD (Centre for Transdisciplinary Development Studies) and NECE-Research Center in Business Sciences, University of Beira Interior, (NECE–UBI).

**Conflicts of Interest:** The authors declare no conflict of interest.

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
