# Peer review of "Information and Public Knowledge of the Potential of Alternative Energies"

_energies, doi:10.3390/en15134928_

Round 1
Reviewer 1 Report
The comments and suggestions are included in the annotated pdf.
While the work is thorough, there are errors in the manuscript that need to be corrected. For instance, instead of "bioethanol", "biotenol" is used.
The citations are not always consistent with the bibliography.
In some cases, the sentences are too long and convey several different ideas. This makes them very difficult to understand. Sometimes the sentences just don't make sense and these are indicated with a question mark.
The main text is in English, so all of the tables must be in English.
There are too many significant figures in the results. Why is standard deviation given to 3 sig figs? One is sufficient.
Line 501 - why would one what to increase dependency and imbalance?
Line 542 - what is the goal of this measure?
Line 600-601 - Bachelor's given twice?
Line 670 - seems to be a disconnect when agriculture is considered a priority, but response to climate change is not. Can the authors elaborate?
Lines 851-2 - Biofuels are not fungible. Would the authors care to elaborate?

Author Response
The comments and suggestions are included in the annotated pdf.
R:|Dear Reviewer, thanks for your great job and enforces for giving me ideas and good advice to improve the paper. Many thanks for all your job in this paper review. Please, see all changes made in blue colour.
While the work is thorough, there are errors in the manuscript that need to be corrected. For instance, instead of "bioethanol", "biotenol" is used.
R:| Corrections were made. Thanks
The citations are not always consistent with the bibliography.
R:| Corrected.
In some cases, the sentences are too long and convey several different ideas. This makes them very difficult to understand. Sometimes the sentences just don't make sense and these are indicated with a question mark.
R:| Improved
The main text is in English, so all of the tables must be in English.
R:| All tables were corrected. Thanks for this warness.
There are too many significant figures in the results. Why is standard deviation given to 3 sig figs? One is sufficient.
R:| Changed
Line 501 - why would one what to increase dependency and imbalance?
Line 542 - what is the goal of this measure?
Line 600-601 - Bachelor's given twice?
Line 670 - seems to be a disconnect when agriculture is considered a priority, but response to climate change is not. Can the authors elaborate?
Lines 851-2 - Biofuels are not fungible. Would the authors care to elaborate?
R:| Please, see all changes made in the lines above
Best Regards and thanks for your great job helping us with paper improvements.
Reviewer 2 Report
The manuscript contains important statistical data but lacks proper scientific writing
standards. The statistical representation needs to be exhaustively analyzed. In this regard,
the manuscript needs a complete rewrite.
1. Overall, writing needs to be polished. There are numerous errors in the use of articles and
grammatical consistencies throughout the manuscript. Here are some of the examples:
a. Line 14: Excessive dependence on oil price--- should be ‘oil’, not price.
b. Unnecessary long sentences throughout the abstract.
c. Line 20: “Biofuels have become popular and have begun to be seen as a valid
alternative to fossil fuels because…”
Please go through the manuscript and revise it accordingly.
2. The abstract should contain the work's motif, presentation of key features of the
methodologies utilized, and some representative results or critical findings. Unfortunately,
the abstract doesn’t meet any of these requirements.
3. What is the novelty of the work? Clearly articulate the limitation of the works available in
the literature and how you are improving upon them in the field.
4. Discussion of the result is extremely limited. Authors must discuss the major findings of
their research in a more elaborative way.
5. The conclusion should not be presented as a part of the introduction. It should provide a
summary of the findings and the motif of the research methodologies.

Author Response
The manuscript contains important statistical data but lacks proper scientific writing standards.
R:|Dear Reviewer, thanks for your great job and enforces for giving me ideas and good advice to improve the paper. Many thanks for all your job in this paper review. Please, see all changes made in blue colour.
Round 2
Reviewer 2 Report
Accept as is.